# Modeling and predicting the overlap of B- and T-cell receptor repertoires in healthy and SARS-CoV-2 infected individuals

**María Ruiz Ortega, Natanael Spisak, Thierry Mora⦿\*, Aleksandra M. Walczak⦿\***

Laboratoire de physique de l'École Normale Supérieure, CNRS, PSL University, Sorbonne Université, and Université de Paris, Paris, France

⦿ These authors contributed equally to this work.
\* thierry.mora@phys.ens.fr (TM); aleksandra.walczak@phys.ens.fr (AMW)

## Abstract

Adaptive immunity's success relies on the extraordinary diversity of protein receptors on B and T cell membranes. Despite this diversity, the existence of public receptors shared by many individuals gives hope for developing population-wide vaccines and therapeutics. Using probabilistic modeling, we show many of these public receptors are shared by chance in healthy individuals. This predictable overlap is driven not only by biases in the random generation process of receptors, as previously reported, but also by their common functional selection. However, the model underestimates sharing between repertoires of individuals infected with SARS-CoV-2, suggesting strong specific antigen-driven convergent selection. We exploit this discrepancy to identify COVID-associated receptors, which we validate against datasets of receptors with known viral specificity. We study their properties in terms of sequence features and network organization, and use them to design an accurate diagnostic tool for predicting SARS-CoV-2 status from repertoire data.

**Data Availability Statement:** The datasets used in this study can be downloaded at the following sources: https://github.com/briney/grp_paper https://clients.adaptivebiotech.com/pub/emerson-

## Author summary

Our immune systems rely on a diverse set of receptor sequences to recognize pathogens. Some of these receptors are found to be shared between many individuals, suggesting they may have been selected to have a functional role. We show that even taking into account selection pressures, the distribution of sequences shared between healthy individuals is completely predicted by chance. However, COVID infections channel people's repertoires and result in an overrepresentation of shared receptor sequences. Using this oversharing, we can identify immune receptors that were involved in clearing Sars-CoV-2 infections and verify them against known SARS-CoV-2-associated receptors. We analyse the properties and the similarities between the identified receptors and propose a statistical repertoire-based tool to identify people who have undergone a COVID infection.

2017-natgen https://clients.adaptivebiotech.com/pub/covid-2020 https://www.ncbi.nlm.nih.gov/bioproject/?term=PRJNA625995 https://www.ncbi.nlm.nih.gov/bioproject/PRJNA645245 https://zenodo.org/record/3886395#.YgWMl33MLQ0.

**Funding:** This work was supported by the Marie Skłodowska-Curie Actions H2020-MSCA-ITN-2017 program no 764698 from the European Union (T. M.), and by a European Research Council Consolidator grant no 724208 (A.M.W.). The salary of M.R.O. was provided in part by the Marie Skłodowska-Curie Actions H2020-MSCA-ITN-2017 program no 764698 from the European Union and the European Research Council Consolidator grant no 724208. The funders had no role in study design, data collection and analysis, decision to publish, or preparation of the manuscript.

**Competing interests:** The authors have declared that no competing interests exist.

## Introduction

The unique antigen receptors expressed on the surface of B and T cells determine the set of epitopes that may be recognized during an immune response. Public sequences shared by many individuals have for a long time solicited potential therapeutic interest [1]. However, many of these receptors are shared by chance [2–5] and not necessarily due to prior antigenic experience. Here, by comparing healthy individuals and COVID-19 patients, we explore the interplay of chance and exposure on T-cell and B-cell receptor sharing.

B-cell receptors (BCR) are composed of a heavy and a light chain, and T-cell receptors (TCR) are made of analogous beta and alpha chains. Each chain is formed via a DNA editing mechanism. This process, called V(D)J recombination, randomly splices genes together from germline segments (V, D and J for heavy and $\beta$ chains, and V and J for light and $\alpha$ chains). In addition, a random number of base pairs are trimmed at the junctions between the segments, and random non-templated ones are inserted. The resulting junctional region makes up the hypervariable complementarity determining region 3 (CDR3), which is key for antigen specificity [6, 7]. After generation, receptors are selected through thymic selection for T cells [8], and central tolerance for B cells [9], to ensure proper receptor functionality and to avoid recognition of self-antigens.

Estimates of the number of possible combinations produced by V(D)J recombination range from $10^{20}$ [10] to $10^{61}$ [11] for TCR, and even more for BCR. These numbers far exceed the number of distinct antigen receptors, $\sim 10^8$–$10^{10}$ [12, 13], or even the total number of B and T cells in the human body, $\sim 10^{12}$ [14]. Despite this large difference, unrelated individuals share a considerable number of public sequences, as observed both in antigen-specific repertoires and in the whole repertoire as revealed by high-throughput sequencing [13, 15, 16].

While the sharing of public sequences was initially attributed to selection biases due to encountered antigens, it was then proposed that public sequences may be explained by convergent recombination [2]. The V(D)J recombination process generates biases that favor certain sequences over others, and those frequent sequences are likely to be found in many individuals.

However convergent recombination itself is not enough to explain the amount of overlap between the TCR repertoires of unrelated individuals [5]. Convergent selection, which amplifies biases in sequence statistics, is further needed to explain it quantitatively. Convergent selection can either stem from peripheral selection triggered by encountered pathogens or from avoiding self-antigens. Learning a single, global sequence-independent correction factor allows one to predict the full spectrum of sharing observed among a large cohort of donors [5]. However, this approach ignores sequence-specific features of selection, and can only imperfectly predict which sequences are public or private. In addition, it has not been applied to B cells for which there is no quantitative theory of repertoire sharing.

Here we develop a detailed framework to explain and quantitatively predict immune receptor sharing through both convergent recombination and convergent selection, as schematized in Fig 1. We integrate selection pressures acting on receptor sequences [17, 18] into a statistical theory of repertoire sharing [5].

We apply this framework to successfully predict the sharing spectrum of TCR and BCR repertoires in human cohorts. By contrasting predictions in cohorts of healthy donors against patients recently infected with COVID-19, we identify lists of "overshared" sequences with putative specificity for SARS-CoV-2, which we validate against existing databases. Our approach not only predicts sharing among healthy individuals, but can also be used to identify antibodies and T-cell receptors of interest that are both specific and public, opening new avenues for applications of repertoire sequencing to diagnostics, precision medicine and vaccine design.

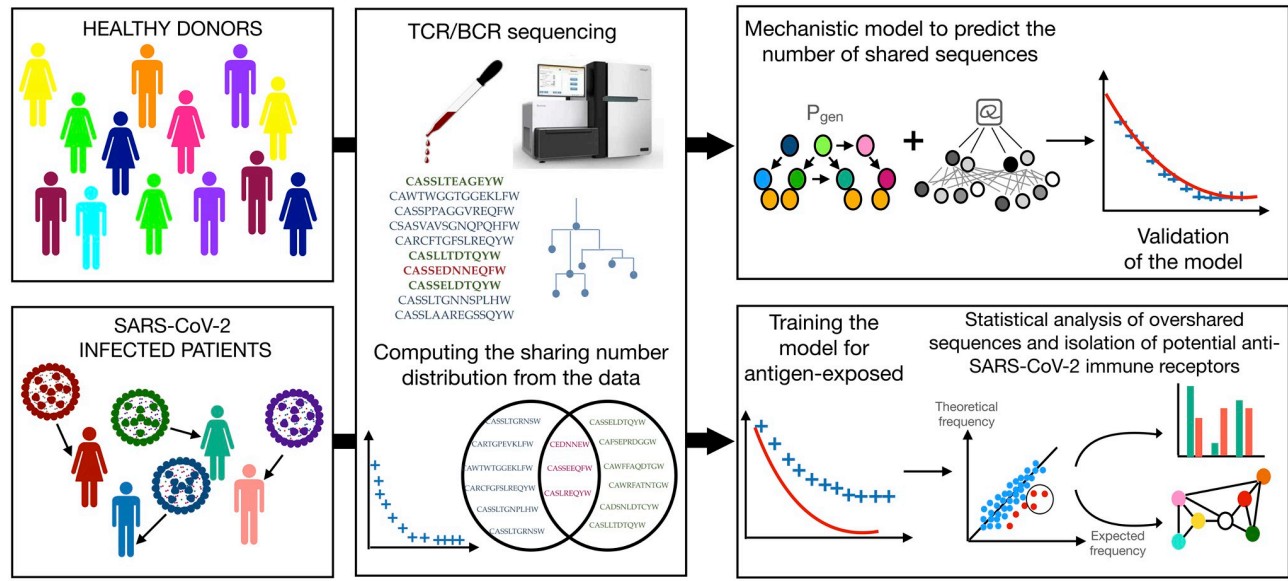

**Fig 1. Schematic of the analysis pipeline.** The BCR or TCR repertoires of a cohort of healthy donors are sequenced from blood samples. Raw reads are processed by removing duplicates and extracting the CDR3. For a given cohort, the distribution of sharing numbers $m$ is obtained by counting the number of sequences that are found in exactly $m$ individuals. In parallel, sequences are used to train a probabilistic generative model, in a two-step process: first learning the recombination model $P_{gen}$ from nonproductive sequences; second learning a selection model $Q$ to describe productive sequences. The model is validated by comparing its prediction for the distribution of sharing numbers with the data. The repertoires of individuals currently or recently infected with SARS-CoV-2 are collected and processed as described for healthy donors. The distribution of sharing numbers is compared with the model prediction obtained previously, this time showing departures from the model, due to the enriched sharing of public COVID-associated clonotypes. These clonotypes are identified as those that are significantly more frequent in the cohort than predicted by the theory. The sequence features and organization of COVID-associated clonotypes are then analyzed and validated against databases of receptors with known SARS-CoV-2 specificity.

## Results

### Predicting sharing in healthy BCR repertoires

We aim to quantify how convergent recombination and selection influence and shape the repertoire landscape. To that end, we examined the degree of sequence overlap across different BCR and TCR repertoires. We first combine all the unique CDR3 amino acid sequences of immunoglobulin heavy chains (IGH) or T cell $\beta$ chains (TCR$\beta$), which we refer to hereafter as "clonotypes," from each individual of the respective cohorts.

For each clonotype, we computed their *sharing number*, i.e. the number of individuals in which it appears. The sharing number offers a nuanced view in which clonotypes are neither purely "public" nor "private", but may instead have different degrees of publicness. The "sharing spectrum" can be visualized by plotting the distribution of sharing numbers, i.e. the number of clonotypes corresponding to each sharing number (Fig 1).

We investigated sharing in IGH repertoires. We analyzed high-throughput sequencing data from a cohort of 10 healthy donors totalling more than $3 \times 10^8$ IGH sequences, separated into IgM and IgG subpopulations [13]. Although the number of people seems quite low, the repertoire size is large enough for the results to be considered robust and trustworthy. Since sharing is dependent on the repertoire size and at present there exist no other repertoires sequenced at such great depth, we would have to downsample these 10 repertoires, also losing generalizability. The resulting sharing number distribution of their IgM repertoires is shown in Fig 2A, (blue crosses). To explore the forces behind the existence of public sequences, we compared the distribution of sharing number in experimentally sampled datasets with theoretical

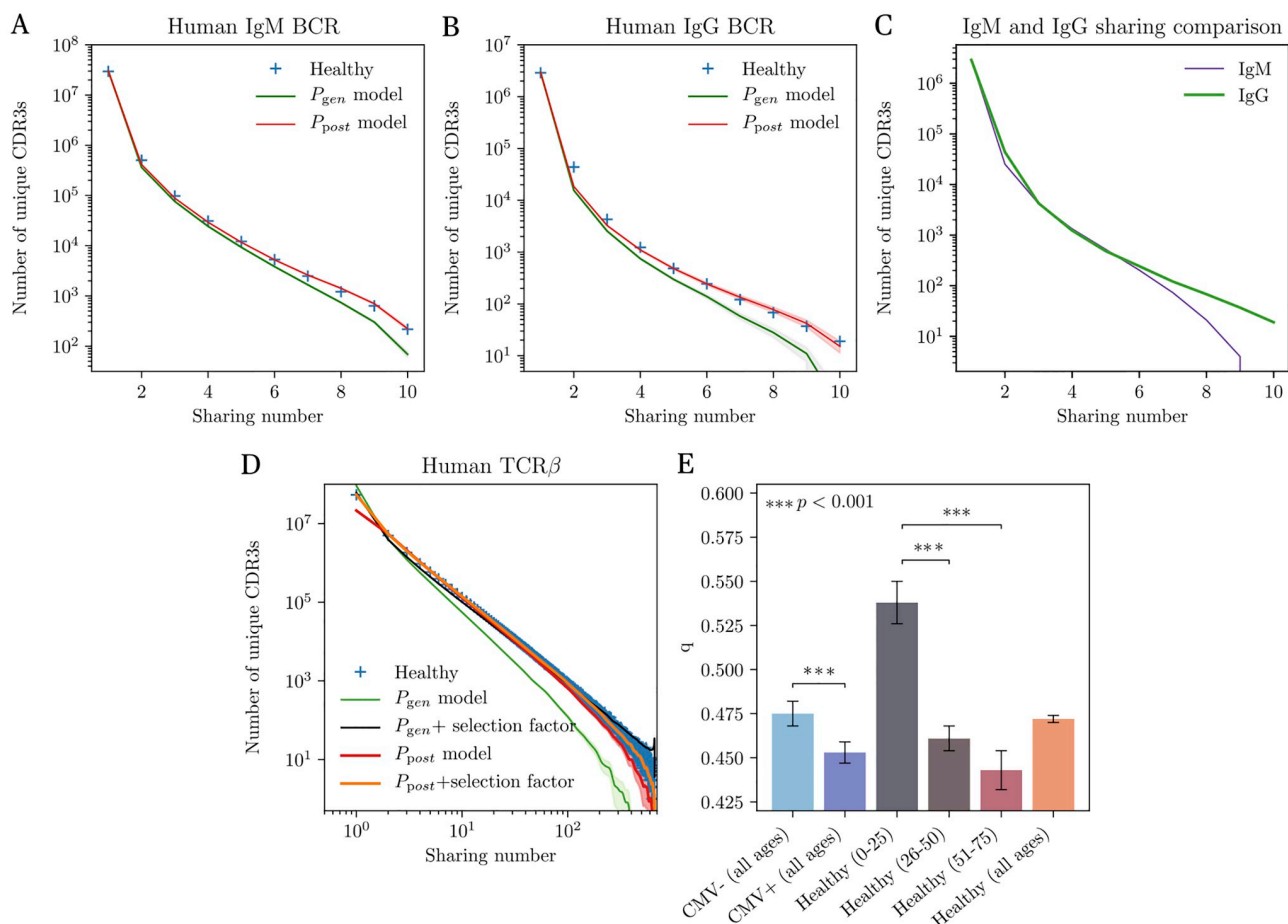

**Fig 2. Sharing of IgH and TCR$\beta$ CDR3 repertoires of healthy individuals.** (A) Distribution of the sharing number (the number of individuals in which a sequence is seen) of CDR3 amino-acid sequences of the heavy chains of IgM repertoires from 10 individuals. The prediction from the raw recombination model ($P_{gen}$, green line) underestimates sharing. Adding an *ad hoc* correction factor assuming a fraction $q$ of sequences passing selection ($q = 0.759 \pm 0.001$) gives a good fit (see S1 Fig). The prediction from the generation and selection model ($P_{post}$, red line) reproduces the curve perfectly, with no need for a correction factor. (B) Distribution of sharing number for the IgG repertoires of the same donors. The analysis is done on the naive ancestors of reconstructed clonal lineages. The $P_{gen}$ model is again inaccurate, requiring a correction factor $q = 0.636 \pm 0.005$, while the $P_{post}$ works well. (C) Comparison of the sharing number distribution between two equal-size cohorts of IgM and IgG repertoires. For an identical sequencing depth, IgG repertoires present a higher level of sharing, suggesting stronger convergent selection than in IgM repertoires. (D) Distribution of the sharing number of CDR3 amino-acid sequences of TCR$\beta$ from 666 patients. Models predictions are shown for $P_{gen}$ and $P_{post}$, with or without a correction factor $q$. The correction factor is $q = 0.037$ for $P_{gen}$, and $q = 0.472 \pm 0.002$ for $P_{post}$, indicating a better accuracy of the latter. In addition, the corrected $P_{gen}$ model (black line) overestimates the number of sequences shared by all individuals relative to the data and to the corrected $P_{post}$ prediction (orange line). (E) Value of the corrective factor $q$, interpreted as the inverse strength of convergent selection ($q = 1$: no selection; $q \ll 1$: strong selection), for different subgroups of the TCR$\beta$ cohort. We observe stronger selection in CMV-positive individuals than in CMV-negative ones, reflecting their common antigenic exposure. Convergent selection also substantially increases with age. An additional control to account for the possible convergent selection bias due to CMV + content in each age group showed negligible influence. Significance obtained with student's t test.

predictions from previously described models for immune receptor sequences. Given a model $P(\sigma)$ for the probability of occurrence of any clonotype $\sigma$, we can predict the sharing spectrum either by simulating synthetic repertoires, or by using the technique of generating functions, which is computationally faster (see [5] and Methods).

The first model we consider describes the generation process in terms of the statistics of the V(D)J recombination process, denoted by $P_{gen}(\sigma)$. It is inferred from the nonproductive nucleotide sequences of the IgM repertoires of all donors, using the IGoR tool [19]. This model

accounts for convergent recombination, which is the most evident driver of sharing. However, $P_{gen}$ alone underestimates the amount of sharing (Fig 2A, green curve obtained from the model prediction and red crosses in S1 Fig obtained directly from sequence simulation). We hypothesized that this discrepancy could be explained by the effects of selection that follow the generation process, which include functionality checks and central tolerance. To test this, we trained a selection model on top of $P_{gen}$, but this time on productive clonotypes, using the SONIA package [20]. Mathematically, the probability of a clonotype $\sigma$ takes the form $P_{post}(\sigma) = Q(\sigma)P_{gen}(\sigma)$, where $Q(\sigma)$ is a sequence-dependent selection factor made up of a product of individual amino-acid choices at each position along the CDR3. We also compared (S7(A) Fig) the performance of SONIA with an algorithm already published, immuneSIM [21], for which we used the available default model. The immuneSIM model could be improved by tuning some of the repertoire parameters, such as germline gene usage, the occurrence of insertions and deletions or clonal abundance, this information must be obtained manually, while in IGoR+SONIA case there was no need for data preprocessing thanks to the automatic learning by IGoR. Additionally, simulation with immuneSIM takes $\sim 33$ mins compared to $\sim 57$ s using SONIA. The resulting sharing distribution from an in silico, same sized BCR dataset shows a clear overestimation for the simulation. $P_{post}(\sigma)$ characterizes a given CDR3 amino acid sequence not only by its probability of being stochastically generated, but also of having passed selection and belonging to the functional repertoire. The prediction of this model for the sharing spectrum shows excellent agreement with the data (Fig 2A, red curve), implying that a combination of convergent recombination and functional selection is enough to explain the overlap between repertoires.

We extended this approach to the analysis of memory IGH repertoires, as captured by the IgG population composed of antigen-experienced cells (Fig 2B). To apply the same method as to IgM populations, we reconstructed the set of putative ancestral sequences of the IgG receptors. These ancestral sequences correspond to the unmutated sequences that cells carried prior to affinity maturation. Since they are rid of hypermutations, they are similar to naive sequences from inexperienced cells. However, they differ from them because they have passed an additional selection step, through their activation and initial recruitment to germinal centers. In practice, the ancestral inference was done by dividing all nucleotide IGH sequences into clonal families, and by reconstructing the most likely naive progenitor of each family (see Methods). We then computed the sharing spectrum of the ancestral clonotypes across the same 10 donors (Fig 2B, blue crosses). We learned an IgG-specific selection model $P_{post}$ from the ancestral clonotypes, on top of the generative model learned above. Similarly to IgM, the model predicted sharing with high accuracy (Fig 2B, red curve), while using the generation model $P_{gen}$ alone underestimated it (green curve). Our statistical approach is thus able to capture the convergent selection not only of naive clonotypes, but also of memory precursor clonotypes. It is important to underline that for repertoires of the same size, the level of sharing observed in this case is even higher than for IgM (Fig 2C). This may be a possible signature of the selection phase occurring during the transition from IgM to IgG, in which high-affinity B cells are positively selected in germinal centers [22].

That analysis only considered the sharing of the CDR3 amino acid sequences, which is the most variable part of the receptor. However, one can use a stricter definition of sharing by imposing V gene and the J gene identity as well. Doing so yields a notable drop in the number of shared clonotypes in both IgM and IgG repertoires (S2(A) and S2(B) Fig), as expected. Nevertheless, the sharing distribution is still well predicted by simulations with this new definition, meaning that our $P_{post}$ model adequately captures the convergence not only of the CDR3 but also of the V and J genes.

## Sharing in TCR repertoires depends on age and CMV status

Next, we applied our framework to TCR repertoires. We trained selection models from the combined productive TCR$\beta$ repertoire of 666 healthy donors, totalling $\approx 1.4 \times 10^7$ unique nucleotide sequences [16], on top of a $P_{gen}(\sigma)$ generation model trained on the nonproductive sequences from the same cohort. We tested two selection models: a SONIA model as before, and a more advanced model based on artificial neural networks implemented in the soNNia package [18]; training was done on a subsample of $10^6$ amino acid sequences. By contrast to B cells, not only the generation model but also both models of selection substantially underestimated sharing (Fig 2D, green and red curves). This discrepancy suggests that, while the selection model predicts clonotype frequency accurately [18], it misses important features of thymic selection, e.g. its dependence on HLA types which is not included in the model.

To overcome this issue, we introduce an additional step of convergent selection: we assume that only a fraction $q$ of sequences sampled from $P_{post}(\sigma)$ is selected. This fraction is picked at random but fixed once for all donors. The single parameter $q$ is learned from fitting the relationship between the number of unique clonotypes and the number of unique nucleotide sequences, which depends on $q$ but is not directly related to the sharing spectrum [5] (see Methods). Fitted values of $q$ close to 1 mean that the selection model $P_{post}$ captures convergent selection well, while smaller values of $q$ imply that additional selection is at work. Accordingly, we found increasing values of $q$ as the model gets more accurate: $q = 0.037$ for $P_{gen}$ (Fig 2D, black curve) [5], $q = 0.072$ for SONIA, and $q = 0.472$ for soNNia (Fig 2D, orange curve). The sharing predictions using this last value of $q$ for soNNia are excellent. The same pipeline was applied using the vampire software [23] to generate sequences, for which we obtained an overestimation of sharing (S7 Fig). An analogous evaluation was made for B cells using SONIA, finding $q = 0.759$ and $q = 0.636$ for the $P_{gen}$ model in IgM and IgG, respectively (black curves in S1(A) and S1(B) Fig).

## Ageing in TCR repertoires

More than a correction factor, $q$ may be interpreted as the stringency of selection. The smaller is the q value, the less diverse will be our repertoire, and the more sharing we expect. We used this interpretation to study repertoire evolution with age. Ageing is believed to be accompanied by a gradual loss of diversity in T cell receptor repertoires [24–26]. This phenomenon is due to a progressive deterioration of thymus functionality [27, 28] together with an expansion of memory T cell clones [29, 30] generated by former encounters with antigens.

To study the influence of age on selection, we split the cohort from [16] into three ages subclasses (0–25, 26–50 and 51–75 years old, with 81, 339 and 134 individuals each, respectively), and inferred a different $q$ factor in each group: $q_{0-25} = 0.538 \pm 0.012$, $q_{26-50} = 0.461 \pm 0.007$ and $q_{51-75} = 0.443 \pm 0.011$. The 0–25 group had a significantly less stringent selection factor than the other two age groups, consistent with repertoires becoming more and more convergent with age (Fig 2E, grey, brown and maroon bars). We further checked that this effect was not due to the increased proportion with the age of donors who tested positive for Cytomegalovirus (CMV). CMV-positive donors share more sequences than CMV-negative donors (light and dark blue bars), because of enhanced convergent selection for CMV specificity among HLA-matched donors [16]. To control for this confounding effect, we used the Fisher-Yates shuffle algorithm to create 3 control groups with no age structure, but with the same respective proportions of CMV+ donors as in the original data. Applying the same pipeline to the control groups yielded statistically indistinguishable values of $q$, meaning that age and not CMV status is the main driver of convergent selection. We also applied this analysis to an additional control dataset [31] for which we observed a similar behavior except for the age group 75–105 (S8

Fig). Given the unknown medical conditions of these people, it is hard to draw conclusions from this increase in diversity but it opens up an interesting path of research.

## Convergent BCR sharing in COVID-19 donors

Having validated the predictive power of the SONIA model for BCR sharing in healthy donors, we asked whether it was still predictive for repertoires stimulated by a common disease such as COVID-19. We studied the IgG heavy-chain repertoires of 44 SARS-CoV-2-positive individuals (as confirmed by RT-qPCR test), obtained from three independent studies [32–34], all collected in 2020 when only the original Wuhan strain was circulating. All data provided in these studies correspond to a post-infection time point from bulk sequencing. Despite the inherent interest in inferring the dynamics of public repertoires, the availability of longitudinal data for the large numbers of people needed for the sharing analysis remains very limited or nonexistent. A similar problem arose when we tried to extend the analysis to single-cell immune profiling data. The cost of single-cell experiments makes it very hard to get the large repertoire sizes needed to get statistically meaningful results. Fig 3A shows that the model trained on healthy patients substantially underestimates sharing between the IgG repertoires of individuals with COVID-19. This enrichment of shared sequences among COVID-19 donors is strongly suggestive of convergent selection for sequences that have common SARS-CoV-2 epitope specificity. This overrepresentation cannot be explained by simple features such as the number of N nucleotides or gene usage, which are already accounted for by the model. The same effect is observed when also imposing V and J identity in the sharing definition (S2(C) Fig).

To identify the putative clonotypes with SARS-CoV-2 specificity that are responsible for oversharing, we first estimated, for each shared clonotype $\sigma$, its frequency $P^*_{\text{data}}(\sigma)$ in the COVID-19 cohort. This estimate is based on the number of donors in which it was found as well as its multiplicity (synonymous nucleotide variants) in each donor (see Methods); this information is gathered in the occurrence vector $\vec{x} = x^\sigma_1, x^\sigma_2, \ldots, x^\sigma_M$ giving the number of nucleotide sequence variants coding for $\sigma$ in each individual 1, . . ., $M$. We expect that number to generally correlate with the model prediction derived from healthy donors, $P_{\text{post}}(\sigma)$, as shown in Fig 3B. However, we also expect COVID-19 specific clonotypes to be over-represented in $P_{\text{data}}$ versus $P_{\text{post}}$, and fall in the lower-right part of Fig 3B. To identify those clonotypes, we first applied a constant multiplicative factor to $P_{\text{post}}$ to correct for the systematic underestimate of $P_{\text{data}}$ (see Methods).

Next, we call "COVID-associated" clonotypes whose estimate of $P_{\text{data}}(\sigma)$ was significantly higher than the rescaled $P_{\text{post}}$, with a false discovery rate of $10^{-4}$ (using Bayesian analysis, see Methods). This procedure identified 6650 COVID-associated clonotypes when using only CDR3 aa identity for the definition of sharing (CDR3 sharing, S1 Table), and 2502 if we impose V and J gene identity (full amino-acid sequence sharing, S2 Table). To validate their COVID specificity, we cross-checked this list against a database of antibodies with reported specificity for SARS-CoV-2, which we built from the literature (compiled in S3 Table) [35–49]. Among the COVID-associated CDR3, 175 sequences ($\approx 2.6\%$ of the overshared sequences) had at least a 90% Levenshtein similarity (one or two mismatches or gaps) in amino acid content with the CDR3 of reported SARS-CoV-2 binding antibodies. Of those, 161 shared the same V gene family and 92 the whole V gene with the sequences from the donors (versus 132 and 32 if we randomly shuffle V genes and CDR3 in the database of SARS-CoV-2 antibodies). We also directly compared our 2502 COVID-associated, full amino-acid, sequences with the database, reporting 30 hits ($\approx 1.2\%$).

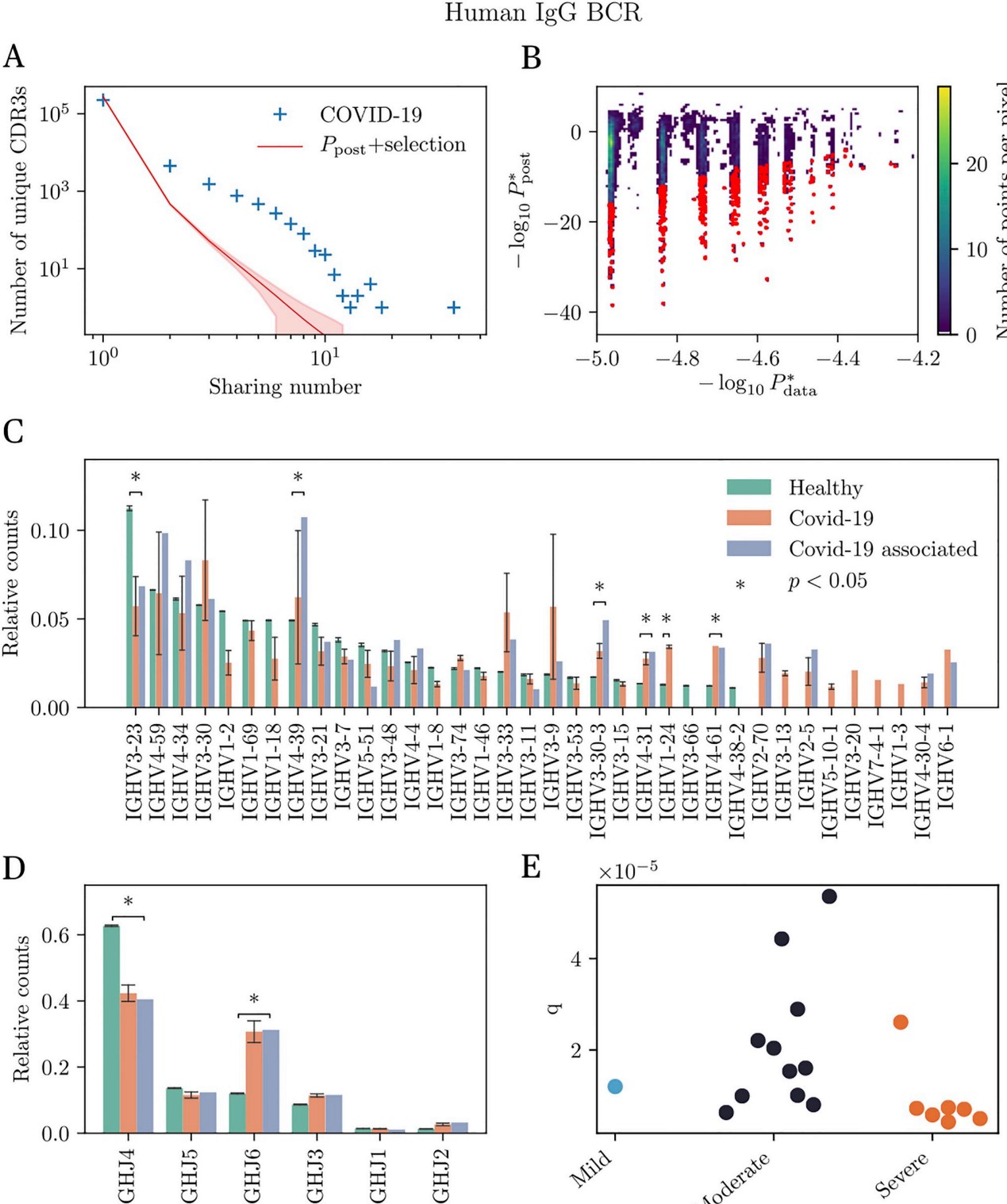

**Fig 3. Identification and analysis of COVID-associated antibody heavy chains from significantly shared sequences.** (A) Sharing number distribution of IgG heavy chain CDR3 from SARS-CoV-2 positive individuals. This distribution is compared with the sharing expectation in healthy individuals, obtained using the $P_{post}$ model. The discrepancy suggests the enrichment of these repertoires in SARS-CoV-2 specific antibodies. (B) Scatter plot showing the sequence probability derived from our theoretical model, $P_{post}^*(\sigma)$, vs. the empirical probability $P_{data}^*(\sigma)$ of all shared sequences. This parameter is obtained by maximum-likelihood estimation and represents the value of the probabilistic model under which the observed data are the

most probable. Red sequences are substantially more frequent in the data than predicted (posterior probability $<10^{-4}$), and are predicted to be associated with SARS-CoV-2. (C) IGHV gene usage in healthy (green) and COVID-19 (orange) repertoires, and among COVID-associated sequences (blue). (D) IGHJ gene usage for the same groups (same color code). (E) Convergent selection factor $q$ learned from pairs of COVID-19 individuals in different severity group. Individuals with more severe symptoms seem to have a higher level of selection, although the difference is not statistically significant (Student's t test ($p = 0.12$))).

To better understand the significance of this overlap, we carried out two control analyses. First, we followed the same pipeline to identify significantly overshared sequences in the IgG healthy cohort, and looked for sequence similarity of these overshared sequences in the assembled SARS-CoV-2 antibody database. We found no similar sequences, confirming the intuition that one should not find SARS-CoV-2 specific antibodies in the repertoires of healthy people sampled before the pandemic. Second, we constructed a mock antibody database of the same size as the original one, by taking random samples from the pool of healthy IgG sequences. Sequence matching among significant sequences in COVID-19 repertoires and this control database yielded an overlap of 2.7 ± 1.5 sequences (0.04%) that are 90% similar by Levenshtein similarity when we look at CDR3 similarity and of 2.2 ± 1.2 (0.09%) when we also impose V and J gene identity. This result is also coherent since we don't expect all overshared sequences to be just due to SARS-CoV-2 infection.

Not all of the 6650 overshared sequences are necessarily COVID-19 specific response, as they can correspond to other common past infections. To test this, we also compared them to sequences in a database [50] containing specific antibodies for many other maladies and infections such as influenza, HIV, allergies or CMV. We obtained 46 matches (versus 0 in a random subsample of non-shared sequences) with a predominance of allergies (10 matches) and Ebola (11 matches). This result suggests convergent selection shaped by other forces than COVID.

We then asked whether the COVID-associated antibodies differed in the CDR3 sequence features from other antibodies found in healthy or COVID repertoires. In Fig 3C and 3D show the comparison of V and J gene usage between COVID-associated and bulk repertoires. We observe a very diverse usage of V and J genes in COVID-associated sequences. While no V or J gene stands out as being COVID-specific, there are a few remarkable differences, such as the enrichment of IGHV3–30-3, IGHV4–31, IGHV1–24, IGHV4–39, IGHV4–61 and IGHJ6 in COVID-associated sequences, whereas IGHV3–23 and IGHJ4 are underrepresented. Note that while some of these genes are already shared in healthy patients, due *e.g.* to long stretches of templated base pairs into the CDR3 as in IGHJ6, as predicted by our model, these differences are more marked in COVID sharing. The CDR3 length distribution (S3(A) Fig) shows a small but substantial bias towards longer sequences in COVID-19 repertoires (orange curve) compared to healthy repertoires (green curve), consistent with previous findings [32–34, 43].

We also examined the hypermutation rates in the three groups. COVID-associated sequences had lower hypermutation rates than healthy repertoires (S3(B) Fig). Since repertoires were taken during the infection, these B cells have expanded quickly but may not have had the time to accumulate mutations compared to mature memory B cells. However, decent neutralization power can be achieved without long affinity maturation [40], a feature already observed in immune responses to other respiratory viruses [51].

Finally, we wondered whether the degree of convergent selection was predictive of disease severity. To answer this question, we focused on the data from [33], which contains information about the severity (mild, moderate, and severe) of each individual. Due to the small number of individuals in the cohort, we opted for a bootstrap method to estimate individual variability. We defined a selection factor $q_{AB}$ for each pair of individuals A and B as the correction needed to exactly predict their repertoire overlap using the $P_{post}$ model as a baseline. The

lower $q_{AB}$, the more convergent the repertoires of A and B are. The resulting $n(n-1)/2$ pairwise selection factors (Fig 3E) show increased convergent selection among individuals with severe forms of the disease, although the data was not sufficiently powered to identify with certainty a statistically significant difference between the groups.

## Network analysis of COVID-specific repertoires

The CDR3 region is the major determinant of the binding specificity of both BCR and TCR [52, 53]. We expect that upon infection, repertoires will be enriched with CDR3 clonotypes with similar sequences recognizing predominant epitopes [54–58]. We looked for motifs among the COVID-associated clonotypes through a sequence-based network analysis. Such clustering analysis was previously used to visualize repertoires from individuals suffering from chronic diseases [59], giving a useful graphical representation of the structural differences among clonally expanded and healthy repertoires.

We calculated Levenshtein distances between all pairs of CDR3 clonotypes and V, J gene identity. The choice of this metric is motivated by simplicity and biological reasons. Considering that B and T cell receptors usually undergo insertions/deletions/substitutions Levenshtein distance becomes a more appropriate measure than Hamming distance, which only works with strings of the same size. Its implementation is also very simple and appropriate for pattern recognition, exactly what we aim for when we look for conserved motifs in CDR3 regions. Clonotypes are considered adjacent if their distance is $\leq 2$. We clustered clonotypes using single-linkage clustering from this adjacency matrix. These clusters correspond to functionally similar sequences, which are not related through a common ancestor and may come from different donors. The lack of paired $\alpha$-$\beta$ sequences stopped us from using more sophisticated software as [60] The resulting cluster are shown in Fig 4A, where each node is a clonotype, and adjacent clonotypes are linked by edges. Sequence logos [61] under each cluster show that CDR3 sequences are well conserved within each group.

To assess the robustness and significance of the clusters, we designed a control cloneset obtained by pooling together the bulk IGH repertoires from all 44 COVID-19 patients. We built 5 random subsets of the same size as the COVID-associated subset, and repeated the network analysis on each. Note that while specificity clusters are still expected in bulk repertoires, they are likely to stand out less than in specific repertoires, where diversity is reduced and sequence similarity higher. Consistent with this, only 3.6±0.2% of sequences of the control subsets belonged to non-singleton clusters, compared to 8.6% for the COVID-associated clonotypes. This implies that the network analysis captures the sequence convergence of functional origin.

One hypothesis is that each cluster corresponds to a distinct specificity group targeting a given epitope associated with SARS-CoV-2. To test this, in Fig 4A we marked clonotypes found to be close to previously reported SARS-CoV-2 antibodies with a color that depends on which antigenic region of SARS-CoV-2 they recognize.

In the largest cluster (labeled 1 in the figure), 13 out of 97 sequences (14%, versus 0.1 ± 0.3% for random clonotypes) are just one amino acid away from antibodies recognizing the receptor binding site (RBD) of the S protein.

Entry into human cells is initiated by interaction between the RBD and the cell surface receptor ACE2 [62–66] making RBD an immunodominant target for neutralizing antibodies [67]. Two clusters (3 and 4) include matches with antibodies interacting with other epitopes than RBD (non-RBD and S2 domain, which contain the fusion peptide). V and J gene usage is well conserved across clusters 2, 4 and 5 (IGHV3–33, IGHV3–30, IGHV3–30-3 and IGHJ6), while central positions of the CDR3s are enriched in different residues, consistent with their

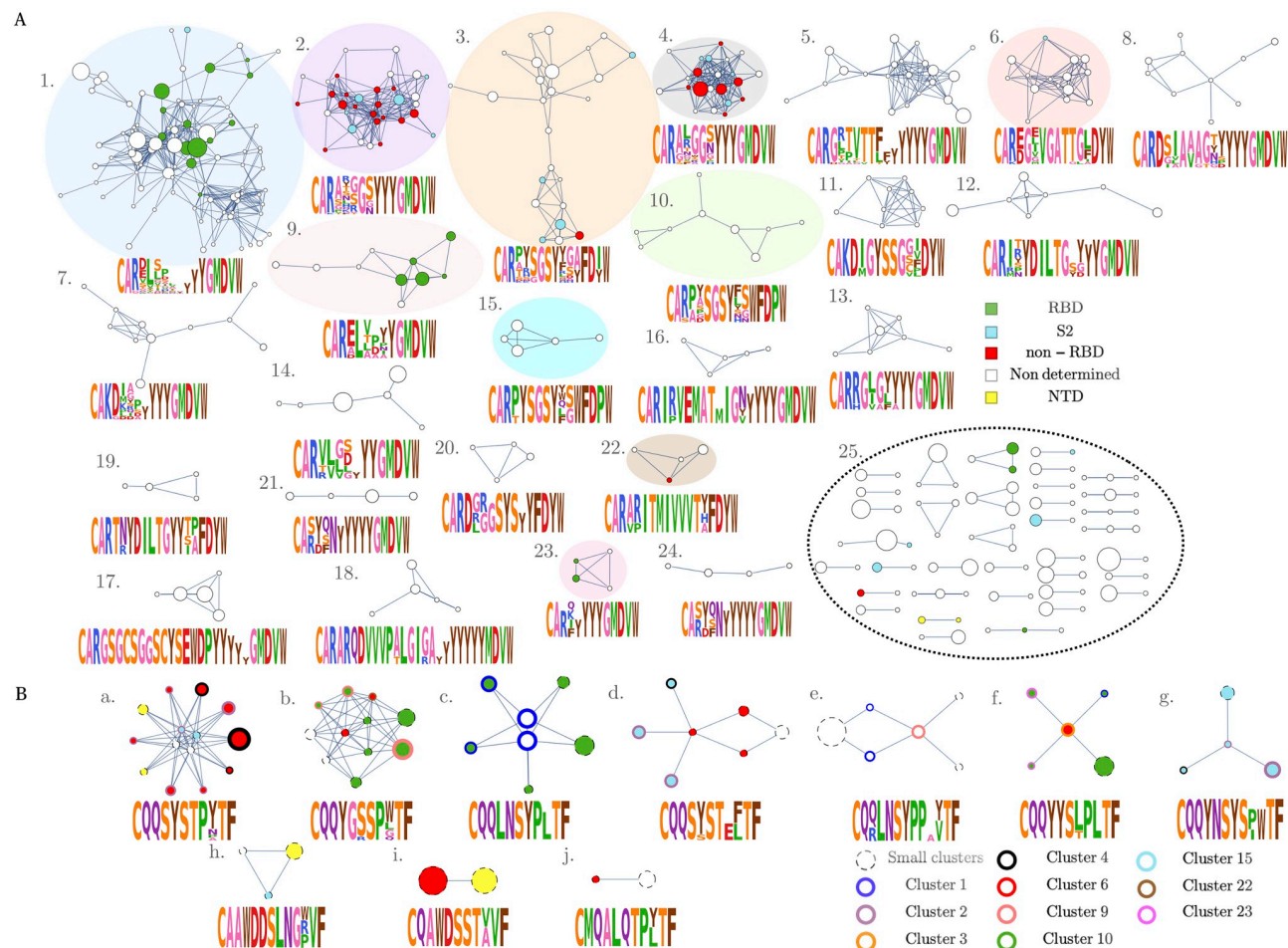

**Fig 4. Networks analysis of IgG heavy chains associated with SARS-CoV-2.** (A) Each node represents a CDR3aa heavy chain clonotype. Edges connect clonotypes with two or fewer amino acid mismatches in their CDR3 region, and with the same V and J segments. Non-connected vertices are not shown. Colored nodes represent sequences that are at most one amino acid mismatch away from previously reported SARS-CoV-2 neutralizing antibodies, with the color indicating the region of the virus recognized by the antibody. The size of the node is proportional to its sharing number, i.e. the number of people in our cohort where the sequence is found. (B) Network of light chains CDR3 amino acid sequences found in previous reports to be paired with a colored heavy chain of (A). The level of conservation is even more remarkable than for heavy chains.

distinct specificities. The non-determined specificity corresponds to sequences for which we didn't find any match with the public antibodies found. These results are very dependent on the available information at the moment. However, since COVID-19 studies are rapidly evolving, we have that those non-determined clusters will be soon identified also as SARS-CoV-2 specific as more and more data are published.

We then asked whether there was an analogous structural selection acting on light chains. While we do not have light-chain repertoire datasets from COVID-19 donors, we can collect the light chains of all SARS-CoV-2-specific antibodies from the literature that matched (up to 2 aminoacid substitutions) our COVID-associated clonotypes in their CDR3. Applying the same clustering procedure as described above yields 10 different clusters, shown in Fig 4B), with the color of the circle outline coding for the cluster membership of the heavy chain of that clone in A. For example, cluster (c) is exclusively associated with anti-RBD antibodies, and matches heavy-chain cluster (1). Cluster (g) represents a clear antigen-specific cluster with all

components being antibodies acting on S2, but does not have a clear analogue among heavy chain clusters, suggesting that specificity is primarily determined by the light chain. For light chains, the level of redundancy observed was even greater than for heavy chains. This was in part due to the fact that some SARS-CoV-2 confirmed antibodies from the literature differ in their CDR3 heavy chain, but have identical light chains, and this may be a general consequence of the light chain being less variable than the heavy chain.

## Predicting sharing in T cell repertoires from COVID-19 patients

T cells play an important role in COVID-19 infections, and can persist for years even when IgG antibodies have become undetectable [68, 69]. We applied our sharing analysis to the TCR$\beta$ repertoires of 1414 donors with confirmed SARS-CoV-2, sampled at various timepoints following the peak of the disease, and totalling $1.6 \cdot 10^8$ reads [70].

Fig 5A shows the sharing spectrum for all 1414 donors. We recall that, by contrast to IGH, a corrective selection factor $q$ was needed to allow the statistical model $P_{\text{post}}$ to correctly predict the sharing spectrum, even for healthy donors (Fig 2D). This selection factor may be interpreted as quantifying the level of convergent selection among donors, as demonstrated for CMV+ donors (Fig 2E). Inferring $q$ on the COVID-19 cohort using the soNNia model yielded $0.452 \pm 0.002$ (Fig 5A, orange curve), compared to $q = 0.472 \pm 0.006$ for the healthy cohort (Fig 5A, red curve), suggesting that COVID-19 donors shared more TCR among them due their common infection. With this slight change, the corresponding prediction for the COVID-19 sharing spectrum is excellent.

While convergent selection is less evident than for B cell repertoires, highly clonal T-cell repertoires have been reported by several studies [43, 71]. Inspired by these results, we used the same approach as for B cells for identifying COVID-associated TCR. In Fig 5B, we compare the soNNia-inferred probabilities $P_{\text{post}}(\sigma)$ with the empirical frequencies of the receptor sequences $P_{\text{data}}(\sigma)$. The potential responding clones marked in red were identified as those whose empirical frequency was unexpectly high relative to the model prediction, as described before for B cells. To focus on clonotypes with the largest significance, we picked the 0.1% clonotypes with the highest posterior probability of being COVID-associated, resulting in a list of 20,841 sequences (S4 Table). To validate the specificity of this list, we compared it to a list of clonotypes with know specificity to SARS-CoV-2 epitopes, as confirmed by a Multiplex Identification of Antigen-Specific (MIRA) assay [70], and found 3,722 (17%) overlapping clonotypes. By contrast, random lists of 20,841 sequences from healthy patients and from COVID-19 associated datasets (drawn from a biased distribution to reflect their sharing properties, see Methods) only shared $27 \pm 5$ and $193 \pm 13$ clonotypes with the MIRA assay, respectively. These much lower quantities demonstrate not only that the cohort of significantly shared sequences is enriched of SARS-CoV-2 specific clones but also indicates preexisting cross-reactive T cells recognizing SARS- CoV-2 epitope in previously unexposed individuals, probably related to other seasonal coronaviruses [71–73].

The analysis of sequence features revealed a slight increase of CDR3 length in COVID-associated versus generic clonotypes (S4 Fig). A few V and J genes were enriched or depleted in COVID-associated TCR (Fig 5C and 5E): TRBV5–1, TRBV20–1 and TRBV6–2, TRBJ1–1 are considerably underrepresented in the COVID-associated repertoire, while TRBV27, TRBJ2–1 and TRBJ1–2 are enriched, consistent with previous reports [43, 74].

To better understand the structure of the COVID-specific TCR repertoire, we next performed a network and clustering analysis of COVID-associated clonotypes. As before, we built a graph by putting edges between clonotypes with at most two differences in CDR3 amino-acid identity or in V and J gene usage. Clonotypes found in the MIRA dataset of [70] are

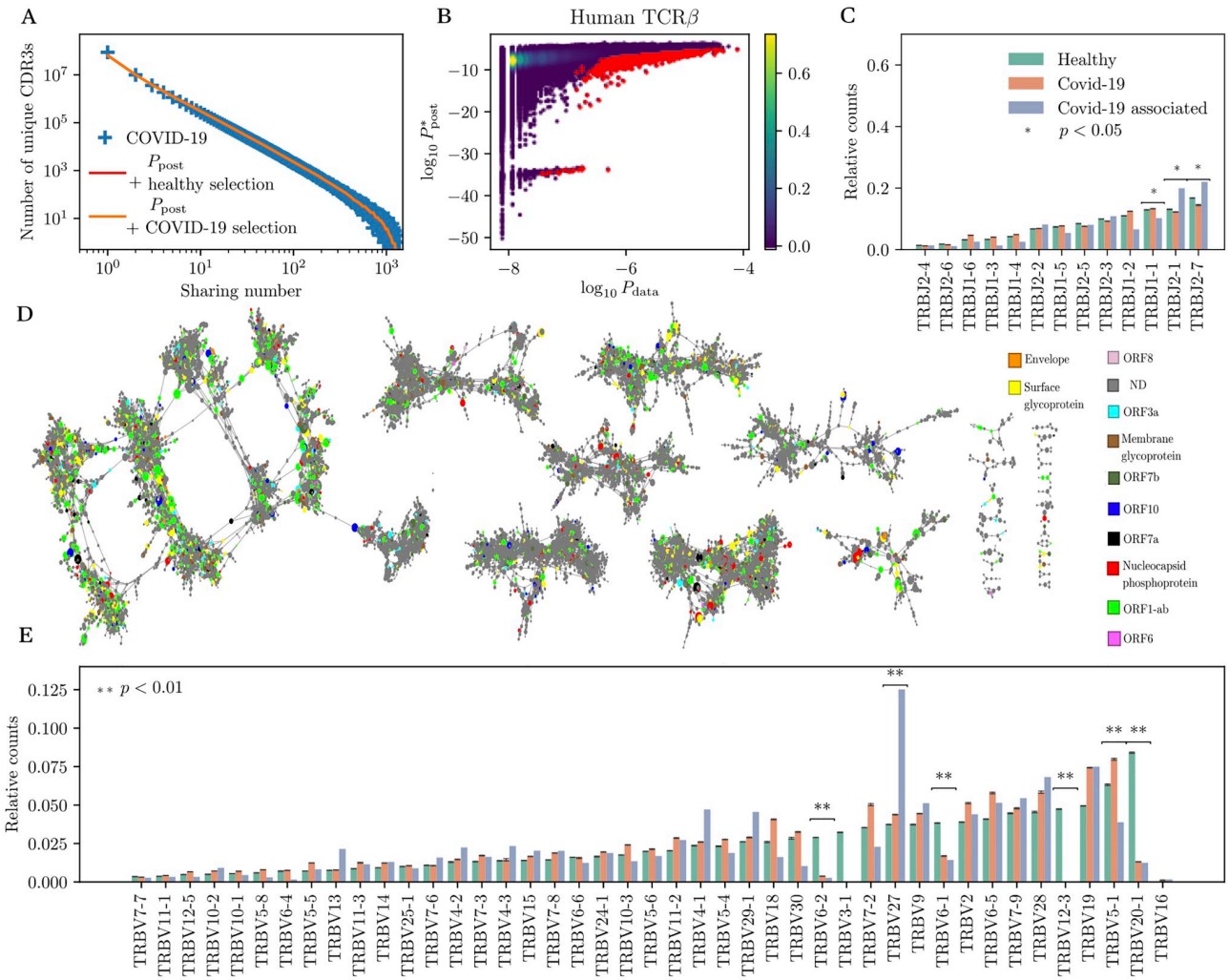

**Fig 5. Identification and analysis COVID-associated TCR$\beta$.** (A) Sharing number distribution of CDR3 amino-acid TCR$\beta$ clonotypes from 1414 SARS-CoV-2 positive individuals. The model prediction from $P_{post}$ (trained on healthy donors, red curve) is good and almost indistinguishable from the one corrected for COVID-19 selection (orange curve), suggesting low convergent selection. (B) Model prediction $P^*_{post}(\sigma)$ versus empirical frequency $P^*_{data}(\sigma)$ for all shared CDR3 amino-acid clonotypes. Clonotypes marked in red are significantly more frequent in COVID-19 donors than expected. (C) TRBJ gene usage in generic healthy and COVID repertoires, and in COVID-associated sequences. (D) Network analysis of TCR$\beta$ amino acid CDR3s of COVID-associated clonotypes. Edge mark clonotypes with 2 mismatches or fewer. Vertices are colored according to the location of their recognized antigen according to the MIRA assay (see main text). (E) TRBV gene usage for the same groups. Significance is obtained using Student's t test.

colored according to their antigen specificity: yellow for the surface glycoprotein (S), red for the nucleocapsid phosphoprotein (N), brown for the transmembrane protein (M) and orange for the envelope protein (E); and the seven putative accessory proteins (ORF1ab (light green), ORF3a (light blue), ORF6 (magenta), ORF7a (black), ORF7b (dark green), ORF8 (light pink), and ORF10 (dark blue). The resulting network and clusters (Fig 5D) paint a more complex picture than for B cells, with no clear sign of grouping by specificity. However, the level of clustering is significant: repeating the same graph analysis of the control set of clonotypes from healthy patients (corrected for their sharing properties as above, see Methods) yields a much lower level of clustering: 12.87 ± 0.13%, versus 88% for the COVID-associated clonotypes. This means that, while clusters are not able to discriminate between different antigenic targets, selection for antigenic specificity still drives a local convergence of clonotypes.

## Using public sequences for SARS-CoV-2 infection diagnosis

The public COVID-associated clonotypes identified above can be used as a biomarker to detect exposure to current and past virus infections. Within our probabilistic framework, we developed a likelihood-ratio classification test based on comparing the likelihood of occurrence of COVID-associated clonotypes in unexposed versus exposed individuals. These likelihoods were computed using $P_{post}(\sigma)$ for unexposed individuals, and the empirical clonotype distribution $P_{data}(\sigma)$ from COVID-19 donors for exposed individuals (see Methods for details).

We first applied this strategy to repertoires of IgG ancestors. We started from the previously identified list of 6650 clonotypes significantly overshared in COVID-19 patients. $P_{data}(\sigma)$ was estimated empirically from the repertoires of 20 COVID-19 donors reserved for training.

Plotting the distributions of log-likelihood ratio scores in SARS-CoV-2 positive and negative individuals held out for testing (20 positives and 10 negatives) shows a clear separation of the two groups across a threshold value of 0, indicating perfect discrimination by this score for this small cohort (Fig 6A). This result implies that this score could be used for the diagnostic of COVID-19.

We next applied the same method to TCR$\beta$ repertoires, using 700 SARS-CoV-2 positive individuals to train $P_{data}$, and keeping an equivalent number for testing. The raw log-likelihood score using the 1,200 top COVID-associated clonotypes yielded poor discrimination power (S5 Fig). To improve performance, we trained a logistic regression classifier predicting COVID-19 status based on the presence or absence of top COVID-associated clonotypes as predictors. The weights of the model were learned using the same training set, and performance was evaluated on the remaining donors as a testing set. Classification error was optimized by tuning the number of included clonotypes and regularization parameters to avoid overfitting (see Methods for details). The distribution of logistic regression scores (expressed as the probability of being positive) among the positive and negative cohorts of the testing set is shown in Fig 6B. Setting a threshold at 0.5 gives 83% specificity and 92% precision. Varying the threshold yields the Receiver Operating Curve shown in Fig 6C, which summarizes the classification performance for both methods (logistic and likelihood-ratio test). The performance of our method is lower than that reported in [70], where a larger cohort was used, and where some clonotypes leading to classification errors were filtered out by hand.

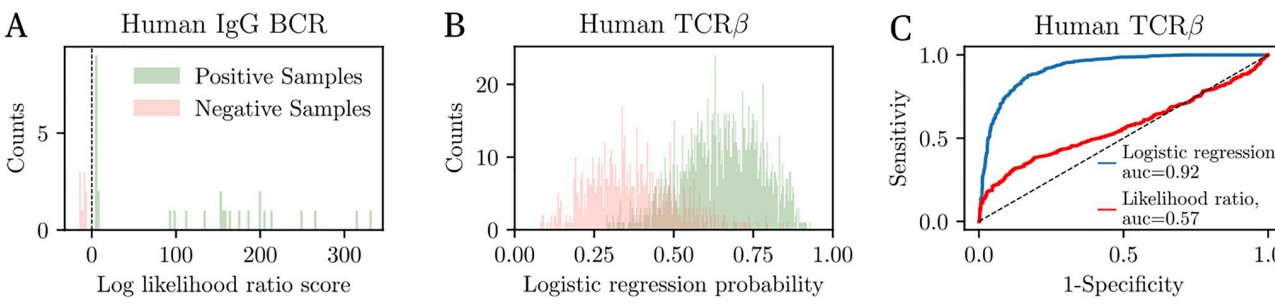

**Fig 6. Repertoire-based SARS-CoV-2 diagnostics.** (A) Distribution of log likelihood ratios calculated from the IgG repertoires of 30 individuals (20 SARS-CoV-2 positive, and 10 negative). Positive values of the score imply likely SARS-CoV-2 positivity. The test perfectly separates positive and negative individuals. (B) Distribution of probabilities of SARS-CoV-2 positivity obtained from logistic regression on the presence or absence of clonotypes in TCR$\beta$ repertoires (1,000 individuals from the testing set; model trained on 1,000 individuals from the training set). Values over 1/2 indicate likely SARS-CoV-2 positivity. The distributions of healthy and COVID-19 individuals have little overlap (93% specificity). (C) ROC curve obtained by tuning the positivity threshold in (B). The AUC is the area under the curve (AUC = 1: perfect discrimination; 1/2: no discrimination). ROC of the likelihood test is shown for comparison.

## Discussion

Our statistical model allowed us to characterize in detail how the generation and initial selection of repertoires could explain the observed spectrum of sharing among unrelated healthy donors. The methodology relies on two ingredients: a stochastic recombination model, $P_{gen}(\sigma)$, and a selection model, $P_{\text{post}} = Q(\sigma) \times P_{\text{gen}}(\sigma)$, where $Q(\sigma)$ is a sequence-specific selection factor. A previous attempt at predicting sharing in TCR [5] only considered $P_{\text{gen}}$. This model alone underestimates sharing, but applying a single corrective selection factor $0 < q \leq 1$ yielded good predictions. Our analysis shows that learning a proper selection model $Q(\sigma)$ from data allows for excellent predictions for BCR sharing, without the need for such a corrective factor. This suggests that for B cells, $P_{\text{post}}$ accurately captures selection effects encoded in peripheral B cells, including central tolerance and affinity maturation. This success is surprising because we could have expected central tolerance to remove auto-reactive B cells, thereby reducing diversity and increasing sharing. This effect would be hard to capture by a linear model such as SONIA, which we used for $Q(\sigma)$.

For T cells, the picture is more nuanced. Using $P_{\text{post}}$ yielded much better predictions than $P_{\text{gen}}$ alone, but a corrective factor $q$ was still needed to get a perfect fit. The value of $q$ is indicative of the level of selection not captured by the model but still necessary to explain sharing. Its increase from $q = 0.037$ when using $P_{\text{gen}}$, to $q = 0.472$ when using $P_{\text{post}}$, is a strong signature of model improvement. A possible hypothesis for why $q$ is still needed is that negative thymic selection reduces diversity and increases sharing. Linear or even smooth non-linear models such as soNNia may not be able to capture such effects. The thymic selection also strongly depends on the HLA type, which determines what peptides may be presented to T-cells during selection, and this information is specific to each individual. One needs to go beyond universal models to make personalized predictions of clonotype prevalence and sharing.

SARS-CoV-2 infection substantially affects the antibody repertoire [75–77]—around 0.15–0.8% of total memory B cells are spike-specific according to [78]. We expect to observe a pool of overshared BCR clonotypes among COVID-19 donors, corresponding to a convergent response. Our pipeline identifies such overshared, antigen-specific receptors. Since these antibodies are both COVID-specific and found in many donors, and they are likely to be elicited in many patients. Such identification of universal antibodies could help in the design of next-generation vaccines, or therapies based on neutralizing antibodies [35, 79–81].

Out of the 236,079 IgG ancestral clonotypes across all 43 COVID-19 donors collected from the literature, 20,113 (8.5%) were shared among at least two individuals. Within that pool, 6,650 were found to be significantly overshared, among which 175 had at least a 90% Levenshtein similarity with previously reported anti-S-protein antibodies. The most shared IgG ancestral clonotype (CARGFDYW) was linked to the Kawasaki syndrome, an inflammatory disease that has been associated with COVID-19 [82]. Although this CDR3 is found to be linked to 25 V distinct genes across donors, the most abundant V association (IGHV4–59, in found in 32 out of 89 sequences), has been repeatedly reported to be found in SARS-CoV-2 infected individuals [83]. Among significantly shared sequences, we also matched some clonotypes with previously described antibodies, such as the CV30 (CARDLDVSGGMDVW, IGHV3–53) in [46], the COVA2–07 (CAREAYGMDVW, IGHV3–53) in [41] and the COV2–2381 (CAAPYCSRTSCHDAFDIW, IGHV1–58) in [63]. All of them were reported to reach complete neutralization through competition with the ACE2 human protein for RBD binding, which is the main mechanism used by anti-SARS-CoV-2 antibodies. Because RBD is likely to accumulate escape mutations, it may be useful to identify antibodies targeting more conserved epitopes of the S protein. The high similarity of the S2 subdomains between SARS-CoV-2 and SARS-CoV

[63] suggests that targetting conserved epitopes may also allow for cross-neutralizing different coronaviruses [84].

Despite a common focus on humoral immunity against COVID-19, T cells play an essential role. The analysis of immunity in recovered SARS-CoV and MERS patients showed that the T-cell response is often more durable, remaining detectable up to 17 years after infection, even when antibodies are undetectable [85]. We applied the same pipeline for identifying over-shared TCR clonotypes in a cohort of 1414 COVID-19 donors. While the level of convergent selection was much less striking than for BCR, the method was still able to detect a subset of 20,841 significantly shared sequences, 17% of which were separately validated for SARS-CoV-2 specificity. We hypothesized several reasons for the absence of convergence in T-cell immune response. First, genetic differences in the HLA genes are known to influence the composition of TCR repertoires which can partially modify the individual response to pathogens [86]. The inclusion of HLA type in future models could improve the search and validation of antigen-specific clonotypes.

It is important to emphasize the versatility of the sharing analysis which, despite its simplicity, can be applied to very different diseases, such as an acute respiratory infection (SARS-CoV-2), a chronic infection (CMV) or an autoimmune disease (as ankylosing spondylitis, previously discussed in [58]). In our specific case, the compatible values of the q factors fitted on each cohort, $q_{CMV} = 0.453 \pm 0.006$, $q_{SARS-CoV-2} = 0.452 \pm 0.002$, suggests a much less dramatic change than for B cell repertoire composition. Distinguishing the public response from other autoimmune diseases or tumor conditions remains to be done for several reasons, like the lack of deep and numerous sequenced repertoires affected by the same immune condition or the requirement of an overall generalized response to a given antigen. It is now widely known that in many cancers, each patient likely has a unique genomic molecular signature, making sharing not the most suitable approach to understanding the immune response in this case [87].

Our results are consistent with IgG repertoire diversity being reduced in patients with severe diagnostic, suggesting that this parameter could help predict the evolution of the disease [88, 89]. Relatedly, we used our selection parameter $q$ to measure the difference in repertoire diversity of healthy donors from three age groups. We found significant differences between the youngest (0–25 y.o.) and the oldest (51–75 y.o.) healthy groups, implying greater T-cell diversity in younger individuals. It has been speculated that efficient protection against COVID-19 in children may arise from their highly diverse T-cell repertoire, while older adults may be at higher risk due to immunosenescence [90]. Further statistical analyses to study the relationship between age, repertoire diversity, and severity could help better predict disease evolution.

In this context of condition-associated sequence analysis, previous studies [70, 71, 91] yielded the identification of pre-existing SARS-CoV-2-reactive TCR clones in unexposed individuals. The role that they might play in asymptomatic or mild COVID-19 cases remains unclear. However, they could serve as a basis for thinking about vaccines eliciting a broader immune response. Since the responding B and T cell clones identified by this method repeatedly appear in different individuals, they constitute an interesting vaccine target since they could trigger a robust public response.

The clinical potential of this method also extends to the development of a B-cell or T-cell repertoire-based diagnostic tool to identify current or past infection from sequenced repertoires. The likelihood ratio test we propose here achieves perfect classification between SARS-CoV-2 positive and negative donors using the repertoires of IgG ancestors. However, the size of the testing cohort is very small (30 individuals), and the method should be validated on larger cohorts for clinical applications.

We speculate that the lower accuracy of the method applied to TCR is partly due to the HLA restriction of SARS-CoV-2 associated clonotypes, reducing their predictive power. However, we showed that the performance could be considerably improved by logistic regression, as was first proposed in [16, 70]. Predictive power might be further increased by including the HLA information of donors, and by using larger cohorts for training. Compared to the method described in [16], we used our trained generative model $P_{\text{post}}(\sigma)$ as a null model, rather than by directly recording clonotype frequencies in the healthy cohort. While this allows for estimating the probability of sequences that were never seen in the healthy cohort, its prediction may be inaccurate for certain sequences, affecting the accuracy of the score. Using a combination of probabilistic modeling for rare clonotypes and empirical frequencies for common ones may provide an intermediate method benefiting from both approaches.

Despite these challenges, developing TCR repertoire-based clinical tests is important and complementary to serological and BCR-based tests, because we expect TCR repertoires to be more stable over time, and COVID-19 signatures to persist detectable in peripheral T cells longer than in antibody sera. More generally, our results are promising for the future of repertoire-based diagnostics. The method is general and applicable to any disease or condition that impacts immune repertoires. As more cohorts of donors with a common condition are screened for associated clonotypes, this approach would allow for doing several tests simultaneously from a single repertoire sample, including retrospectively as more tests are being designed.

## Methods

### Sequence preprocessing

**IgM and IgG repertoires.**   The analysis of healthy BCR heavy-chain (IGH) repertoires was performed on several large datasets comprising a total of over $3 \times 10^8$ productive sequences from 10 healthy adult subjects [13]. The sequence isotype, IgM or IgG, was identified during primer removal and pairing of the two reads. We used the pre-processed data provided by the initial analysis of the dataset [13] with the exception of the two largest individuals (326651 and 326713). For these two donors, we applied a custom pipeline to find high-quality out-of-frame transcripts required for inference of the generative model of V(D)J recombination. Most out-of-frame sequences are expected to originate from a frameshift during recombination, rather than from hypermutations or sequencing errors, insertions and deletions. Those occur with rates lower than $10^{-4}$ per base pair [92] or with a frequency lower than 0.5% in the CDR3, compared to the total 5.6% of unproductive sequences. We aligned raw reads using pRESTO of the Immcantation environment [93] with a setup allowing for correcting for errors in UMIs and dealing with insufficient UMI diversity. Since reads begin with the CDR1, 74% of the V gene and the entirety of the J gene are covered, leading to unambiguous gene assignment. We grouped reads with the same UMI together, annotating each consensus sequence with a corresponding number of reads, which is a proxy for sequence quality. Pre-processed data were then aligned to V, D and J templates from IMGT [94] database using IgBlast [95].

We filtered the set of productive IgM sequences to keep only sequences with at most one mutation. Using a subset of non-productive, naive IgM sequences of high quality (with at least 3 reads per consensus sequence), we used IGoR [19] to infer the statistics of the generative process, and to build a model $P_{\text{gen}}(\sigma)$ that can be used to generate synthetic sequences with no mutations, and free of selection effects that affect real productive sequences.

To include selection effects, we learned a sequence-specific selection factor $Q(\sigma)$ that modifies the probability of occurrences of sequences relative to the generation model. The resulting post-selection probability model reads: $P_{\text{post}}(\sigma) = Q(\sigma)P_{\text{gen}}(\sigma)$. The training of the function $Q$

($\sigma$) was done using IgM productive sequences with at most 1 hypermutation using the SONIA software [20].

**Clonal lineage reconstruction in IgG repertoires.** A large fraction of the IgG repertoire has already undergone successive stages of differentiation and specialization after antigen activation, meaning that statistics describing naive IgM sequences features are no longer valid for IgG sequences. We incorporate these effects in a selection model inferred from the sequence features of the reconstructed naive progenitors of IgG sequences.

To reconstruct IgG clonal families and infer the ancestral sequences, we started by grouping together sequences with the same V and J genes and CDR3 length. Inside each group, we performed single linkage clustering of CDR3 sequences with a threshold of 90% identity [93]. Clusters obtained in this way correspond to clonal families, which we assume originate from a common naive unmutated progenitor. We reconstructed the CDR3 sequences of the naive ancestor by removing mutations from the germline segments of the sequence (using IMGT templates), and taking the consensus sequence for the non-templated segments.

An analogous procedure was used to process IgG heavy-chain antibody sequences from repertoires of 43 COVID-19 donors obtained from [32–34].

**TCR$\beta$ repertoires.** Healthy T cell $\beta$-chain repertoires were taken from a cohort of 666 donors [16], 641 of which were serotyped for cytomegalovirus (CMV): 352 subjects were CMV negative (CMV−) and 289 were CMV positive (CMV+), with a mean number of unique $\beta$ chains $\approx$ 180,000 per donor.

COVID-19 repertoires were obtained from a cohort of 1414 SARS-CoV-2 positive donors sequenced by Adaptive Biotechnologies, as described in [70]. Both datasets were already pre-processed.

We separated non-productive and productive sequences from healthy donors. We inferred a generative model from the nonproductive sequences using IGoR [19], and a selection model from the productive sequences using soNNia [18], an artificial neural network based extension of SONIA.

## Analytic prediction of sharing from the density of probabilities

Given $M$ repertoires sampled from $M$ individuals, each with a number $N_i$ of unique nucleotide sequences, $i = 1, \ldots, N$, the expected number of sequences sharing among exactly $m$ individuals is given by the coefficients of the generating function [5]:

$$G(x, \{N_i\}) \approx \int_0^{+\infty} \rho(p)dp \prod_{i=1}^{n} [e^{-N_i p} + (1 - e^{-N_i p})x], \tag{1}$$

where $\rho(p)dp$ is the total number of potential sequences whose probability falls between $p$ and $p + dp$. Given a model distribution $P(\sigma)$, the integral in 1 is performed by Monte-Carlo: generate a large number of sequences from $P(\sigma)$, evaluate their probability $p = P(\sigma)$, and create a histogram of $\ln p$. Then approximate the integral over $p$ by the method of trapezes.

The simplest model to consider is the generative model $P_{\text{gen}}$ learned from nonproductive sequences, free of any selection. Then Eq 1 is used with $P(\sigma) = P_{\text{gen}}(\sigma)/f$ calculated using OLGA [96], where $f$ is the probability that the generative model produces a productive sequence (given by the IGoR model).

As a minimal to include thymic selection, and in particular negative selection that removes self-reactive receptors from the repertoire, we can introduce an *ad hoc* selection factor $q$ representing the fraction of receptors that are not self-reactive. Then the probability distribution of

non-self-reactive productive sequences is

$$P(\sigma) = \frac{P_{\text{gen}}(\sigma)}{fq},\tag{2}$$

for a random (but fixed across individuals) fraction $q$ of sequences, and $P(\sigma) = 0$ for all the others. This amounts to rescaling the distribution $\rho(p)$ in Eq 1 by a factor $(fq)^{-1}$ in the $p$ variable.

The stronger the selection (the lower $q$), the smaller the number of unique amino-acid sequences will be as a function of the number of unique nucleotide sequences. We can use that relationship to fit the parameter $q$ using least-square regression. The model prediction for the expected number of unique sequences among $N$ random draws from $p$ is given by:

$$M_0^{\text{nt}}(N, q) \approx \int_0^{+\infty} \rho(p) dp (1 - e^{-Np}),\tag{3}$$

where $\rho(p)$ is as before. We can compute that number for nucleotide and amino-acid sequences, $M_0^{\text{nt}}(N, q)$ and $M_0^{\text{aa}}(N, q)$, using the densities $\rho_{\text{nt}}(p)$ and $\rho_{\text{aa}}(p)$ corresponding to these two cases, as described before. The probability of generated sequences, both at the nucleotide and amino-acid level, is computed using OLGA [96] and Eq 2. We then find the $q$ that minimizes the total distance between each pair $\{M_i^{\text{nt}}, M_i^{\text{aa}}\}$ from the data to the parametric curve $M_0^{\text{nt}}(N)$ and $M_0^{\text{aa}}(N)$:

$$q^* = \arg \min_q \sum_{i=1}^{M} \min_{N_i} \left[ (M_0^{\text{nt}}(N_i, q) - M_i^{\text{nt}})^2 + \right.$$
$$\left. + (M_0^{\text{aa}}(N_i, q) - M_i^{\text{aa}})^2 \right]\tag{4}$$

Another way to include selection is to use the sequence-specific selection model $Q(\sigma)$ learned using SONIA or soNNia, and use Eq 1 with $P(\sigma) = P_{\text{post}}(\sigma) = Q(\sigma)P_{\text{gen}}(\sigma)$ instead of $P_{\text{gen}}(\sigma)/)fq)$.

Finally, the two ways to model selection—sequence specific through $Q(\sigma)$, and ad hoc using $q$—may be combined by starting from $P_{\text{post}}$ as a generative model, but assuming that only an unknown fraction $q$ of sequences drawn from it actually survives. This leads to applying the same procedure as in the paragraph above, but with $p(\sigma) = P_{\text{post}}(\sigma)/q$, and $q$ inferred as in Eq 4 but with predictions from the SONIA or soNNia model.

To define a pair-specific selection factor $q_{AB}$ between two individuals A and B for the severity analysis, we fitted the prediction of Eq 1 for the number of sequences shared between two individuals ($M = 2$), using $p(\sigma) = P_{\text{post}}(\sigma)/q$. The fit was done using a grid search on $q$. This factor $q_{AB}$ provides an inverse sharing index between A and B that is robust to the sample size.

## Detecting significant levels of sharing among COVID-19 repertoires

To identify potential COVID-associated receptor clonotypes from oversharing in the cohort of infected individuals, we followed the approach developed in [97], by looking for clonotypes that are more shared in the COVID cohort than expected.

To estimate the baseline probability of CDR3 amino-acid sequences, we applied the SONIA (for B cells) or soNNia (for T cells) model to calculate $P_{\text{post}}(\sigma)$.

We then computed the empirical frequency of sequences in the COVID-19 cohort, $P_{\text{data}}(\sigma)$, from the counts $x_i$ of CDR3 amino-acid clonotype $\sigma$ in individual $i$, but in a probabilistic manner to represent our uncertainty. The probability of observing a receptor $\sigma$ in repertoire $i$ with $\approx N_i$ recombination events (taken to be almost equal to the number of unique nucleotide

sequences) is binomially distributed, so that

$$\mathbb{P}(x_1, \ldots, x_n \mid P_{\text{data}}) =$$

$$= \prod_{i=1}^{n} \binom{N_i}{x_i} [(1 - P_{\text{data}})^{N_i}]^{x_i} [1 - (1 - P_{\text{data}})^{N_i}]^{N_i - x_i}. \tag{5}$$

The maximum likelihood estimate of $P_{\text{data}}(\sigma)$ reads:

$$P_{\text{data}}^* = \underset{P_{\text{data}}}{\arg\min} \, \mathbb{P}(x_1, \ldots, x_n \mid P_{\text{data}}). \tag{6}$$

However in general we will work with the posterior distribution of $P_{\text{data}}$, given by:

$$\rho(P_{\text{data}} \mid x_1, \ldots, x_n) =$$

$$= \frac{\mathbb{P}(x_1, \ldots, x_n \mid P_{\text{data}})}{\int_0^1 \mathbb{P}(x_1, \ldots, x_n \mid P_{\text{data}}) dP_{\text{data}}}, \tag{7}$$

where we have assumed a flat prior for $P_{\text{data}}$.

In practice, we observe a systematic fold-difference between $P_{\text{post}}$ and $P_{\text{data}}^*$. To correct for this effect, we first rescaled $P_{\text{post}}$, $P_{\text{post}}^*(\sigma) = \alpha P_{\text{post}}(\sigma)$. The correction factor $\alpha$ was fitted by minimizing $\sum_{\sigma \in \text{shared}} (\log P_{\text{data}}^*(\sigma) - \log P_{\text{post}}(\sigma) - \log \alpha)^2$.

To evaluate whether a sequence is more present than expected, we compute the posterior probability that its empirical frequency is larger than expected, $P_{\text{data}} > P_{\text{post}}^*$:

$$\mathbb{P}(P_{\text{data}} < P_{\text{post}}^*) = \int_0^{P_{\text{post}}^*} \rho(P_{\text{data}} \mid x_1, \ldots, x_n) dP_{\text{data}}. \tag{8}$$

If this quantity is low, then we have confidence that the sequence is more shared than expected in the cohort, and is therefore called a COVID-associated clonotype. We used different thresholds depending on the context, often conservatively keeping the top sequences, rather than applying a fixed threshold.

## Control cohort with matching $P_{\text{post}}$ distribution

When comparing COVID-associated TCR$\beta$ to sequences from bulk repertoires, we observed differences in their $P_{\text{post}}$ distribution reflecting the relationship between overlap and $P_{\text{post}}$. To make comparisons without that possibly confounding factor, we built a healthy control by histogramming the log $P_{\text{post}}$ distribution of significantly shared sequences in COVID-19 patients into $n$ equal-width bins. For each bin of width $w$ and height $h$, we randomly drew clonotypes from healthy repertoires falling in that $P_{\text{post}}$-bin, until we reached frequency $h$. The resulting dataset has the same distribution of $P_{\text{post}}$ as SARS-CoV-2 sequences, but with sequences from the control repertoire.

## Repertoire-based diagnosis

We used two methods to exploit information about the presence and absence of COVID-associated clonotypes in the repertoire to predict COVID-19 status: a likelihood ratio test, and logistic regression.

The likelihood ratio test evaluates the likelihood of the data under two competing hypotheses: infected with SARS-CoV-2 ($H_1$), or not infected ($H_0$). The log likelihood ratio score then

reads $\mathcal{L} = \log(\mathbb{P}(\text{data}|H_1)/\mathbb{P}(\text{data}|H_0))$, with:

$$\mathbb{P}(\text{data}|H_0) \quad = \prod_{\sigma \in i \cap S}(1 - e^{-N_i P^*_{\text{post}}(\sigma)})\prod_{\sigma \in S \setminus i} e^{-N_i P^*_{\text{post}}(\sigma)}, \tag{9}$$

$$\mathbb{P}(\text{data}|H_1) \quad = \prod_{\sigma \in i \cap S}(1 - e^{-N_i P^*_{\text{data}}(\sigma)})\prod_{\sigma \in S \setminus i} e^{-N_i P^*_{\text{data}}(\sigma)}, \tag{10}$$

where $\sigma \in i \cap S$ means that the sequence is COVID-associated and is found in individual $i$, and $\sigma \in S \setminus i$ that it is COVID-associated but not present in $i$. We took the corrected model $P^*_{\text{post}}$ for the null (uninfected) hypothesis, and the maximum likelihood estimator of the sequence frequency in COVID-19 donors, $P^*_{\text{data}}$, for the infected hypothesis. To avoid overfitting, we learn $P_{\text{data}}$ on a training set.

The set $S$ can be adjusted by taking the $n$ top clones (ranked from lowest to highest value of the score of Eq 8), and letting $n$ vary. Individuals with score $\mathcal{L} > c$ are called positives, and $\mathcal{L} < c$ negative, with $c$ an adjustable threshold, which is set to achieve a given sensitivity on the training dataset.

For the purpose of TCR we turned to logistic regression, which uses a more general form of the score given by the logistic function:

$$\mathcal{L}^i_{\text{logit}} = \log\frac{P(H_1|i)}{1 - P(H_1|i)} = \beta_0 + \sum_{\sigma \in S}\beta_\sigma x^i_\sigma \tag{11}$$

where $x^i_\sigma \in \{0, 1\}$ denotes the absence of presence of sequence $\sigma$ from $S$ in individual $i$. Note that the likelihood ratio test is a particular case, with $\beta_\sigma = \log(e^{N_i P^*_{\text{data}}(\sigma)} - 1) - \log(e^{N_i P^*_{\text{post}}(\sigma)} - 1)$. By constrast, in logistic regression the parameters $\beta_\sigma$ are trained on labeled data, i.e. repertoires of individuals with know COVID-19 status, denotes by $y_i = 1$ if the person was SARS-CoV-2 positive, and $y_i = 0$ otherwise. Logistic regression is performed by minimizing the cross-entropy over the $\beta_\sigma$:

$$\mathcal{H} \quad = \sum_i[y_i \log P(H_1|i) + (1 - y_i)\log(1 - P(H_1|i))]$$
$$+ C^{-1}\sum_\sigma|\beta_\sigma|, \tag{12}$$

where the last term is an $\ell_1$ regularization. The parameters $n$ (controling the size of $S$) and $C$ were optimized to optimize performance (see S6 Fig). With $n = 7000$ and $C = 0.0005$, the logistic classifier achieved 83% specificity and 92% precision on held-out TCR. The dataset was separated between training and testing sets as described in the main text.

## Supporting information

**S1 Fig. Model prediction for CDR3 sharing in the BCR and TCR repertoires of healthy individuals.** (A) Distribution of the sharing number (the number of individuals in which a sequence is seen) of CDR3 amino-acid sequences of the heavy chains of IgM repertoires from 10 individuals. The prediction from the raw recombination model (green line corresponds to model predictions using the $P_{\text{gen}}$ distribution while red crosses are drawn from sequence simulation using the $P_{\text{gen}}$ model) underestimates sharing. Adding an *ad hoc* correction factor assuming a fraction $q$ of sequences passing selection ($q = 0.759 \pm 0.001$) gives a good fit to the data (black curve). The prediction from the generation and selection models ($P_{\text{post}}$, red line) reproduces the curve perfectly, with no need for a correction factor. (B) Distribution of sharing

number for the IgG repertoires of the same donors. The analysis is done on the naive ancestors of reconstructed clonal lineages. The $P_{gen}$ model is again inaccurate, requiring a correction factor $q = 0.636 \pm 0.005$, while $P_{post}$ works well. As one can observe, the selection factor decreases its value from IgM to IgG and since the smaller the q value the stronger the selection, this parameter turns out to be quantifying the selection that accompanies class switching.
(TIFF)

**S2 Fig. Model prediction for full amino-acid sequence (CDR3+VJ) sharing in BCR repertoires.** (A) Distribution of the sharing numberof CDR3 + V gene + J gene amino-acid sequences of the heavy chains of IgM repertoires from 10 individuals. Even though using this more stringent definition of sharing the number of public clonotypes remarkably decreases, the prediction from the generation and selection models ($P_{post}$, red line) still reproduces the curve perfectly. (B) Distribution of sharing number (CDR3 + V gene + J gene) for the IgG repertoires of the same healthy donors and model predictions from $P_{post}$ model simulations. (C) Distribution of sharing number (CDR3 + V gene + J gene) for the IgG repertoires of the SARS-CoV-2 infected donors and model predictions from $P_{post}$ model simulations.
(TIFF)

**S3 Fig. CDR3 length and number of mutations comparison for B cells repertoires.** (A) CDRH3 length distribution averaged over individuals within each cohort of B cell receptors. A control was added for sequences found to be significantly more shared than expected in healthy individuals (named as healthy significant, blue curve), to account for the bias that shared sequences tend to have a shorter CDR3. (B) Violin plots representing the proportion of mutated nucleotides in B cell receptor repertoires per individual in healthy and COVID-19 cohorts, as well as in significantly shared sequences. The dot represents the median SHM percentage. Student's t test: $^{*}p < 0.05$.
(TIFF)

**S4 Fig. CDR3 length distribution comparison for T cells repertoires.** (A) CDRH3 length distributions of generic and significantly overshared TCR sequences. Significantly overshared sequences in healthy individuals (healthy significant, blue curve) are much shorter than significantly overshared sequences in COVID-19 individuals (COVID-19 associated, purple curve).
(TIFF)

**S5 Fig. Distribution of log-likelihood ratio scores for T-cell repertoires.** Distribution of log-likelihood ratio scores for 1000 T-cell repertoires belonging to the test cohort of 700 COVID-19 patients and 300 healthy individuals. The distribution of scores for each group completely overlaps, leading to a classifier with a very poor accuracy.
(TIFF)

**S6 Fig. Parameter selection for the logistic regression model applied to T cell repertoires.** (A) Variation of the logistic regression performance as a function of the total number of sequences. In the absence of regularization, both precision and accuracy reach their maximum for n = 7000 sequences. (B) For the optimal number of sequences found, the precision and specificity are maximized for C = 0.0008.
(TIFF)

**S7 Fig. Methods comparison for sharing prediction.** (A) Distribution of the sharing number of CDR3 amino-acid sequences of IgG repertoires from 10 individuals. The prediction from the model (green line) comes from sequences simulated using the immuneSIM software [21]. The simulation dramatically diverges from the real amount of shared sequences indicating either a bad recombination model or an overestimation of selection. (B) Distribution of the

sharing number of CDR3 amino-acid sequences of TCR$\beta$ from 666 patients. Model prediction (green line) was obtained using the default model of vampire [23] trained on the same data set. vampire slightly overestimates the sharing distribution.
(TIFF)

**S8 Fig. Repertoire diversity in a complementary healthy control cohort.** Values of the selection parameter $q$ as a function of age in repertoires of an independent cohort from [31]. Again, diversity significantly (p<0.001) decreases in repertoires of older people, supporting the idea that aging is accompanied by a loss of repertoire diversity, with the exception of individuals of age 75+.
(TIFF)

**S1 Table. Table of top COVID19-associated CDR3 IgH, containing their CDR3 amino acid sequence, V and J gene choices, $P_{post}$ and $P^*_{data}$ value, posterior probability of being associated ($p$), and index of reported antibody match from the curated database (index column in S3 Table).**
(TSV)

**S2 Table. Table of top COVID19-associated full amino-acid IgH clonotype (CDR3+VJ), containing their CDR3 amino acid sequence, V and J gene choices, $P_{post}$ and $P^*_{data}$ value, posterior probability of being associated ($p$), and index of reported antibody match from the curated database (index column in S3 Table).**
(TSV)

**S3 Table. Table of top COVID19-associated TCR CDR3, containing their CDR3 amino acid sequence, V and J gene choices, $P_{post}$ and $P^*_{data}$ value, posterior probability of being associated ($p$). All reported responding clones come from MIRA dataset.**
(TSV)

**S4 Table. Summary table containing all reported antibodies extracted from the different studies (indicated in column *References*) used for sequence matching. When known, both heavy and light chain information are provided (CDR3 amino acid sequence, V and J gene choices) as well as the antigen target by the antibody.**
(TSV)

## Author Contributions

**Conceptualization:** María Ruiz Ortega, Thierry Mora, Aleksandra M. Walczak.

**Data curation:** María Ruiz Ortega, Natanael Spisak.

**Formal analysis:** María Ruiz Ortega, Thierry Mora, Aleksandra M. Walczak.

**Funding acquisition:** Thierry Mora, Aleksandra M. Walczak.

**Investigation:** María Ruiz Ortega, Thierry Mora, Aleksandra M. Walczak.

**Methodology:** María Ruiz Ortega, Natanael Spisak, Thierry Mora, Aleksandra M. Walczak.

**Project administration:** Thierry Mora, Aleksandra M. Walczak.

**Resources:** María Ruiz Ortega, Thierry Mora, Aleksandra M. Walczak.

**Software:** María Ruiz Ortega, Thierry Mora, Aleksandra M. Walczak.

**Supervision:** Thierry Mora, Aleksandra M. Walczak.

**Validation:** María Ruiz Ortega.

**Visualization:** Thierry Mora.

**Writing – original draft:** María Ruiz Ortega, Thierry Mora, Aleksandra M. Walczak.

**Writing – review & editing:** María Ruiz Ortega, Thierry Mora, Aleksandra M. Walczak.

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
