## [Decision Letter · Decision Letter 0]

30 Nov 2022

Dear Dr Walczak,

Thank you very much for submitting your Research Article entitled 'Modeling and predicting the overlap of B- and T-cell receptor repertoires in healthy and SARS-CoV-2 infected individuals' to PLOS Genetics.

The manuscript was fully evaluated at the editorial level and by independent peer reviewers. The reviewers appreciated the impact of your work, but raised substantial concerns about the current manuscript. Based on the reviews, we will not be able to accept this version of the manuscript, but we would be willing to review a much-revised version. We cannot, of course, promise publication at that time.

If you decide to revise the manuscript for further consideration at PLOS Genetics, please aim to resubmit within the next 60 days, unless it will take extra time to address the concerns of the reviewers, in which case we would appreciate an expected resubmission date by email to plosgenetics@plos.org.

Please do not hesitate to contact us if you have any concerns or questions.

Yours sincerely,

Mark J Cameron, PhD

Guest Editor

PLOS Genetics

Scott Williams

Section Editor

PLOS Genetics

Reviewer's Responses to Questions

**Comments to the Authors:**

Reviewer #1: In this work, Ortega and colleagues analyse overlap of TCR/BCR repertoires among S-CoV-2 neg and pos individuals using statistical models.

I don't have many comments. The main comment is: how does this work conceptually differ from previous works of the research group? The research group has recently published on both Covid repertoires as well statistical models for generation and sharing of immune receptors. What exactly is the novelty in this work as compared to the group's previous research as well as to other works in this space. Can this be made more clear?

For example, this sentence from the abstract "Yet many of these public receptors are shared by chance. We present a statistical approach, defined in terms of a probabilistic V(D)J recombination model enhanced by a selection factor, that describes repertoire diversity and predicts with high accuracy the spectrum of repertoire overlap in healthy individuals. " sounds very similar to previous manuscripts from this group. Please reformat the manuscript to isolate the advance made in this manuscript. Or, if there is no conceptual advance made here, rewrite to focus on the biological insight gained by the data analysis (but also there explain how it is novel compared to prior work in this space).

Minor:

Figure 1 looks a bit disorganized and could benefit from streamlining.

Reviewer #2: In their manuscript ”Modeling and predicting the overlap of B- and T-cell receptor repertoires in healthy and SARS-CoV-2 infected individuals” Dr. Ruiz Ortega et al. describes bioinformatic processes to investigate public B and T cell repertoires, an approach that was also applied to studies of immune repertoires identified in subjects infected with SARS-CoV-2. They provide a convincing analysis framework, but one that also, as other pipelines designed to interrogate big data, requires further assessment.

1. In several instances the focus of the study is entirely on CDR3. This is a common approach in particular in studies of T cell repertoires. CDR3 is certainly at the centre of determination of specificity but it operates in the context of other sequences, such as those encoded by the IGHV. The light chain is of course also a key feature of factors that determine specificity but, as the authors rightfully describe, it is currently not economically feasible to collect sufficient single cell data to carry out a study like this. However, the authors ought to take in particular sequences encoded by IGHV into account for instance in studies of SARS-CoV-2-specific antibodies.

2. In some instances, the authors do to take sequences encoded by the V gene into consideration. Annotation of such data is complicated by the fact that the TRBV and IGHV loci are highly variable in terms of structural variation, gene duplication and the existence of highly similar alleles in different gene locations. How does the analytical approach deal with these aspects, for instance but not restricted to genes like IGHV3-30/3-30-3/3-30-5/3-33?

3. The authors identify a set of non-productive sequences that are used for some comparisons. These reads might have an origin in non-productive transcripts but may also be a consequence of PCR or sequencing errors. How does the authors deal with these matters to ensure that the reads are truly non-functional in the B cell population and not technical artefacts?

4. Public clonotypes seems often to be associated with IGHJ6 (see for instance Figure 3D, Figure 4 and Suppl Fig 2A). This might be a technical artifact. The alleles of IGHJ6 adds a particularly long stretch of residues into CDR3, thereby allowing the for a higher similarity score in the bioinformatic pipeline that allows sequences to reach the identity cutoff. The authors must comment on this.

5. In Figure 4 the authors show CDR3s that differ in length. These differences might certainly be a consequence of insertion/deletion hypermutation events but are more likely a consequence of differential V-DJ or D-J splicing events that exploit similar D/J genes. The frequency of some of the length differences suggests that they do not represent insertion/deletion events. Differences in length of for instance clones 1, 5, 7, 9, 14, 16, 17, 18, 20, 21, 23, 24 are indicative of such artifacts. That puts the true clonal relationship of these groups into questions. Are they really clonally related or just similar clones that happens to represent common sequences. It is not obvious to me if they were derived each from a single subject or derived from multiple individuals; please clarify.

6. Sequences that carry long IGHJ genes but also those that incorporate long IGHD might be better at passing thresholds of the analysis pipeline as N nucleotides represents a smaller fraction of the CDR3. Do the authors identify such aspects of the analysis (for instance, are incorporated D genes of public clones longer than those typically found in rearrangements as such a finding identifies a limitation of the pipeline)?

7. In Figure 4 it is stated that light chain CDR3 conservation is even more remarkable that that of heavy chains. This is entirely expected as these CDR3 show much less diversity in general and depend only on the V-J rearrangement and a limited incorporation of N nucleotides. Human TRBV rearrangements use only a very limited set of D genes but more extensive V gene trimming (as compared to IGHV). How does this affect analysis?

8. CDR3 regions similar to those of SARS-CoV-2 specific clones in an antibody database have been identified (page 7/32). Do these sequences also share V gene? Even if CDR3 is an important part of generation of the specificity it is commonly dependent on other features of the antibodies sequence. We cannot in this context relate to the light chain but the authors would be able to relate to the V-gene.

9. On page 7 (line 227 the authors state (2.7±1.5 sequences). What does it represent in relative terms?

10. In several instances including in Figure 3 and in the text the authors state that they observe a difference but that it is not statistically significant. It would be much better to say something like “the data was not sufficiently powered to with certainty identify a statistically significant difference between the groups”.

11. Are all data sets used full length or do they cover only a part of the IGHV and TRBV genes? If the latter is the case, this may affect gene assignment and clonal binning. Please comment how this may affect analysis and results.

12. Public antibodies have been identified in antibodies against the S-protein / RBD of SARS-CoV-2. Many of the public clones recognize early strains of the virus but not omicron. The present data sets, were they generated from samples collected during the early phase of the pandemic or following circulating of mutated viruses? If some data sets were collected at a point in time when other strains were dominant, this may impact the outcome of analysis. Please discuss how this relates to the results.

13. On line 455 the authors relate to a particular, common CDR3. This sequence has been generated using very little (if any) incorporation of a D gene and limited N nucleotide addition. I can myself identify this sequence in IgM data sets in association with different IGHV genes like IGHV1-2 and IGHV3-74. It is thus not unexpected that it is identified. It may however not relate to one but several clonotypes. Please discuss or delete this particular part of the discussion.

14. The authors may consider shortening the manuscripts as it is quite extensive, e.g. but not limited to the paragraph starting on line 503.

**Have all data underlying the figures and results presented in the manuscript been provided?**

Reviewer #1: Yes

Reviewer #2: Yes

---

## [Decision Letter · Decision Letter 1]

2 Feb 2023

Dear Dr Walczak,

We are pleased to inform you that your manuscript entitled "Modeling and predicting the overlap of B- and T-cell receptor repertoires in healthy and SARS-CoV-2 infected individuals" has been editorially accepted for publication in PLOS Genetics. Congratulations!

Yours sincerely,

Mark J Cameron, PhD

Guest Editor

PLOS Genetics

Scott Williams

Section Editor

PLOS Genetics

Comments from the reviewers (if applicable):

Reviewer's Responses to Questions

**Comments to the Authors:**

Reviewer #1: The authors have address all my concerns.

Reviewer #2: Thank you for providing an extensively updated manuscript though. I noted though that the authors decided not to try to reduce the manuscript in length, probably a disadvantage to the readers.

**Have all data underlying the figures and results presented in the manuscript been provided?**

Reviewer #1: Yes

Reviewer #2: Yes

PLOS authors have the option to publish the peer review history of their article (what does this mean?). If published, this will include your full peer review and any attached files.

Reviewer #1: No

Reviewer #2: No

**Data Deposition**

http://datadryad.org/submit?journalID=pgenetics&manu=PGENETICS-D-22-01187R1

**Press Queries**

---

## [Editor Report · Acceptance letter]

22 Feb 2023

PGENETICS-D-22-01187R1 

Modeling and predicting the overlap of B- and T-cell receptor repertoires in healthy and SARS-CoV-2 infected individuals 

Dear Dr Walczak, 

We are pleased to inform you that your manuscript entitled "Modeling and predicting the overlap of B- and T-cell receptor repertoires in healthy and SARS-CoV-2 infected individuals" has been formally accepted for publication in PLOS Genetics! Your manuscript is now with our production department and you will be notified of the publication date in due course.

With kind regards,

Anita Estes

PLOS Genetics

On behalf of:
